

# Responses of an oyster host (*Crassostrea virginica*) and its protozoan parasite (*Perkinsus marinus*) to increasing air temperature

Jennafer C. Malek[*] and James E. Byers[*]

Odum School of Ecology, University of Georgia, Athens, GA, United States of America
[*] These authors contributed equally to this work.

Corresponding author
Jennafer C. Malek,
malekjc1@gmail.com

## ABSTRACT

**Background**. Changes in climate are predicted to influence parasite and pathogen infection patterns in terrestrial and marine environments. Increases in temperature in particular may greatly alter biological processes, such as host-parasite interactions. For example, parasites could differentially benefit from increased reproduction and transmission or hosts could benefit from elevated immune responses that may mediate or even eliminate infections. In the southeastern United States, the Eastern oyster, *Crassostrea virginica*, is infected by the lethal protozoan parasite, *Perkinsus marinus*. Under field conditions, intertidal (air-exposed) oysters have been found to have significantly higher *P. marinus* infection intensity and marginally higher infection prevalence than subtidal (submerged) oysters. During summer, air temperatures are much warmer than water and this exposure of intertidal oysters to higher temperatures is a suggested mechanism for increased infection intensity.
**Methods**. We simulated intertidal exposure using controlled laboratory experiments to determine how host traits (survival and immune response) and parasite infection intensity will respond to elevated air temperature ranging from 27 °C to 53 °C during emersion at low tide. In Georgia, where our work was conducted, the average summer water temperature is 29 °C and the average maximum high air temperature in July is 33 °C (though oysters have been shown to survive at much higher air temperatures).
**Results**. Host survival declined as temperature increased, with a definitive drop-off between 39–43 °C. Negative effects of air temperature on host immune response (phagocytic activity) were detectable only at extremely high temperatures (47–50 °C) when hosts were suffering acute mortality. Parasite infection intensity peaked at 35 °C.
**Discussion**. Our results suggest that an increase in average summer air temperature to 35 °C or higher could affect oyster survival directly through temperature-related impacts in the short-term and indirectly through increased *P. marinus* infection intensity over the long-term.

## INTRODUCTION

Anthropogenically driven changes in climate are influencing terrestrial and marine ecosystems, from individual species to entire communities (*Janzen, 1994*; *Crick & Sparks, 1999*; *Walther et al., 2002*) Air and water temperature, seawater acidification, and sea level are some of the environmental factors projected to increase due to climate change (*IPCC, 2014a*). Climate models predict that global mean surface air temperature will increase anywhere from 0.3 °C to upwards of 4.5 °C by 2090, with more extreme region-specific increases expected (*IPCC, 2014a*; *IPCC, 2014b*). Such changes in temperature will likely drive considerable variation in biological processes such as metabolic rate (*Gilloly et al., 2001*; *Ganser, Newton & Haro, 2015*) and immune function. For example, temperature can alter both innate and adaptive immune functions including phagocytic and phenoloxidase activity (*Cheng et al., 2004*; *Wang et al., 2008*) and the expression of acquired immune genes (*Raffel et al., 2006*; *Dittmar et al., 2014*). Stimulation or suppression of these important biological processes caused by changes in temperature can affect organismal responses to other factors.

One of the primary uncertainties about the expected shifts in temperature is how such changes might alter interspecific interactions, such as host-parasite interactions. If parasites respond more favorably to increases in temperature than their hosts, through higher parasite growth and reproduction, a warmer world may be a sicker world (*Harvell et al., 2002*; *Harvell et al., 2004*; *Lafferty, Porter & Ford, 2004*). Some parasites have already demonstrated positive responses to climate-induced increases in temperature, including increased geographic ranges (*Pounds et al., 2006*; *Ford & Smolowitz, 2007*) and increased transmission between hosts (*Mouritsen & Jensen, 1997*; *Moore, Robbins & Friedman, 2000*; *Poulin, 2006*). In the Caribbean, for instance, increases in both the prevalence (proportion of infected individuals in a population) and growth rate of yellow band disease lesions in corals co-varied significantly with increases in annual mean water temperature from 1999 to 2007 (*Harvell et al., 2009*). Such positive parasite responses to increasing temperature may be connected to temperature-induced changes in host characteristics, including increased physiological stress (*Lesser & Farrell, 2004*; *Allen et al., 2010*), suppressed immune responses (*Chisholm & Smith, 1994*; *Cheng et al., 2004*; *Al-Zahraa, 2008*), or overstimulated and harmful immune responses (*Wang et al., 2012*).

Alternatively, increasing temperature may benefit hosts more than parasites if the host responds more favorably physiologically than its parasites (*Gehman, Hall & Byers, 2018*). For example, coral and gorgonian hosts have demonstrated stimulated immune responses when exposed to increasing temperature (*Ward, Kim & Harvell, 2007*; *Mydlarz et al., 2008*; *Mydlarz et al., 2009*). *Cheng et al. (2004)* observed that Taiwan abalone had significantly higher respiratory burst activity (rapid release of reactive oxygen species) when exposed to higher (32 °C) compared to control (28 °C) water temperatures. *Mydlarz et al. (2009)* found that the coral *Montastraea faveolata* had higher prophenoloxidase activity (an important precursor to other immune responses used in parasite resistance) when exposed to higher than normal water temperature. Increasing temperature can also stimulate amoebocyte production, and subsequently melanosome production, in

sea fans which concurrently help fight fungal parasite infection (*Mydlarz et al., 2008*). Though a range of host immune responses have been found to be stimulated by increasing temperature, the larger implications of these effects are still uncertain. Because effects of increasing temperature on host immune response, and thus host-parasite interactions, can be varied and are often based on observational studies, we sought to experimentally examine how both host physiological traits and corresponding parasite infection intensity will be influenced by increasing temperature.

Along the East and Gulf of Mexico coasts of the US, the eastern oyster, *Crassostrea virginica*, is an ecologically and economically important ecosystem engineer that creates complex reef habitat and supports a commercial fishery (*Bahr & Lanier, 1981*, *Byers et al., 2015*). Throughout most of its range, *C. virginica* is infected by the protistan parasite *Perkinsus marinus*, which can cause considerable host mortality. First discovered in the Gulf of Mexico in the 1950s (*Mackin, Owen & Collier, 1950*), *P. marinus* has expanded its range from the mid-Atlantic to the Northeast in the last several decades, primarily due to climate-driven increases in winter water temperatures (*Ford & Chintala, 2006*; *Ford & Smolowitz, 2007*). Environmental drivers of *P. marinus* infection patterns are numerous, including water temperature, salinity, diel-cycling hypoxia, tidal elevation, and weather events such as El Niño/Southern Oscillation (ENSO) and the North Atlantic Oscillation (NAO) (*Ford & Tripp, 1996*; *Soniat et al., 2006*; *Soniat et al., 2009*; *Breitburg et al., 2015*; *Keppel et al., 2015*; *Malek & Byers, 2017*). Additionally, the activity and function of oyster hemocytes, which are the primary line of oyster immune defense, are also affected by water temperature, salinity, and diel-cycling hypoxia (*Hégaret, Wikfors & Soudant, 2003*; *Keppel, 2014*). Thus, characteristics of both host and parasite in this system are affected by factors that are predicted to be altered by climate change.

Many of the environmental drivers of both *P. marinus* infection patterns and *C. virginica* immune response are related to temperature. *Malek & Byers (2017)* observed that intertidal oysters had significantly higher intensity (abundance of infection within an individual) and tended to have higher prevalence of *P. marinus* infections than subtidal oysters. They suggested that the mechanism driving this pattern could be exposure to higher mean temperatures or higher variability of conditions, including air temperature, during periods of emersion. In this study, we further investigated the role of air temperature in the *C. virginica*—*P. marinus* system. Specifically, we evaluated if increasing air temperature, including temperatures that are likely to occur with climate change, affect *C. virginica* survival and immune response, as well as *P. marinus* infection intensity.

## MATERIALS AND METHODS

### Oyster collection, preparation, and maintenance

To achieve a more comprehensive understanding of the effects of increasing air temperature on host survival and immune response, we measured these variables in both the presence and absence of *P. marinus* infection. In early May 2014 we analyzed the intertidal oyster population at Priest Landing, Savannah, Georgia, USA ($31°57'44.2''$N, $81°00'46.0''$W) for *P. marinus* prevalence. Using a destructive quantitative PCR assessment (qPCR, see

Parasite Assessment below) we measured a baseline prevalence of 60% ($n = 28$). Using this prevalence, we assumed that oysters collected from this population would provide roughly equal numbers of infected and uninfected hosts (we were unable to determine infection status of individuals until the conclusion of experiment due to the destructive nature of parasite assessment). We randomly collected 225 individual oysters (30–100 mm shell height) from intertidal reefs in late May and cleaned them of sediment and fouling organisms such as juvenile oysters. In the lab we tagged individuals with unique bee tags (The Bee Works), recorded shell heights, and allowed 72 h of acclimation to water conditions of 27 °C prior to beginning air temperature treatments.

To prevent *P. marinus* transmission during the experiment, we held oysters individually in independent 946 ml containers with labels that corresponded to oyster ID tags. We filled containers with artificial seawater (salinity ~25 psu, Instant Ocean dissolved in tap water) and changed the water every 2–3 d to limit accumulation of feces and nutrients. Each container had its own air supply run through a hole in the container lid, with average dissolved oxygen concentrations ranging from 2.9–4 mg/L. We fed each oyster daily with 1 ml of a 2:500 ml dilution of Shellfish Diet 1,800 in deionized water (Reed Mariculture, 2 billion cells ml$^{-1}$).

## General experimental design elements
### Temperature treatments
To determine if host survival and immune response, and parasite infection intensity are affected by increasing air temperature, we manipulated air temperatures to eight values ranging from 27–53 °C at 3–4 °C intervals. This range bracketed both average ambient water temperature (29 °C) and average maximum air temperature (33 °C, with occasional spikes above 40 °C; see Data S1) in Georgia during July, as well as higher air temperatures that oysters could experience based on climate change scenarios (*IPCC, 2014b*). Previous observations indicate that intertidal oysters in the Gulf of Mexico can withstand temperatures from 44–49.5 °C for 3 h (*Ingle et al., 1971*). To ensure that we exceeded this thermal threshold, we also included temperatures above this range, up to 53 °C. A continuously submerged subtidal treatment was held at a constant water temperature of 27 °C and acted as a procedural control. This treatment was used for informal comparison purposes only and was not included in the formal analyses.

We constructed experimental chambers (1 per temperature treatment) using 114 liter plastic bins (Sterilite) wrapped on the exterior with fiberglass insulation to better maintain and stabilize target air temperatures. We transferred oysters from their water-filled individual holding containers to dry air temperature chambers for 4.5 h per day to simulate intertidal exposure, 6 d wk$^{-1}$. The 4.5 h duration was within the range of intertidal conditions experienced in Southeastern US estuaries (*Byers et al., 2015*). Temperature chamber bottoms were lined with sand to create more natural heat absorption conditions. We suspended a clamp lamp with an infrared heat bulb ranging from 75-150 watts from a hole in the center of the chamber lid to heat the chambers to target temperatures. In the bottom corner of each heat chamber we put a temperature sensor (Dallas OneWire DS18B20 Maxim) which was wired to an Arduino Micro micro-controller attached to a

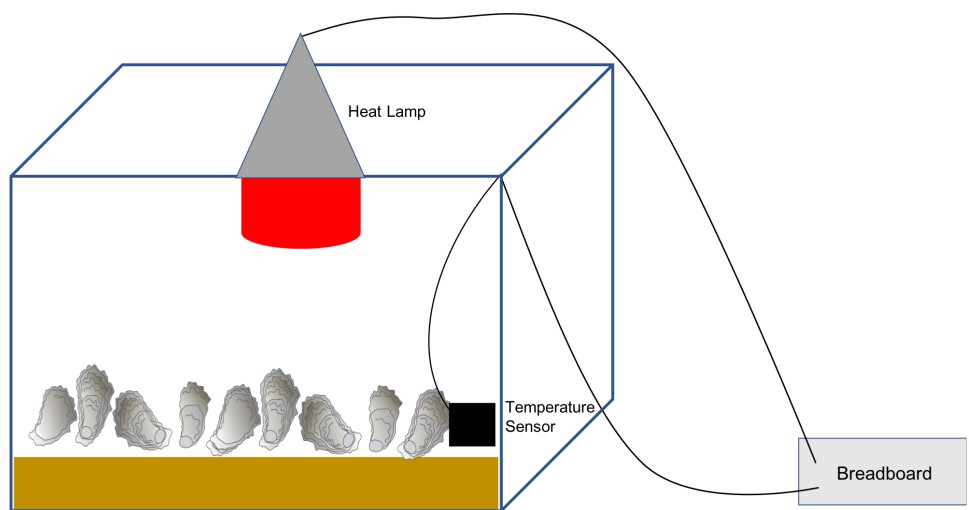

**Figure 1  Experimental chamber setup.** A 114 liter plastic bin was lined with sand and had a heat lamp suspended through a hole in the lid (red circle). A temperature sensor was placed in the chamber just above the surface of the sand (small black rectangle). Both the heat lamp and the temperature sensor were wired to an Arduino Micro microcontroller attached to a breadboard (rectangle to right of the chamber). The heat lamp was turned on and off through the Arduino based on the temperatures relayed back by the sensor. This system kept the chamber within ±0.5 °C of the target temperature.

breadboard. We controlled the power supply of chamber lamps using 8A solid state relays which were also wired to the Arduino through the breadboard (Fig. 1). Code was developed to keep the chambers within ±0.5 °C of the target temperature by turning lamps on and off via the relays in response to the temperatures reported by the sensors.

## Host survival experiment

To develop survival curves for the 25 oysters exposed to each air temperature treatment, we recorded oyster mortality daily prior to starting air temperature exposure and upon return of oysters to their individual containers. Dead oysters were easy to identify from having loose shells or gaping, even when out of water. We collected tissue for *P. marinus* assessment from oysters as soon as mortality was observed. As the parasite does not die when the host dies, we were confident that fresh dead oysters would provide reliable intensity assessment. The experiment ran for 3 weeks and we sampled all remaining live oysters for *P. marinus* infection at the conclusion of the experiment. Thus, we knew how long each oyster survived and its infection intensity upon death or at the end of the three week experiment.

We assessed *P. marinus* infection status and intensity using the Ray's Fluid Thioglycollate Medium (RFTM) method that detects live *P. marinus* spores in oyster tissue. Because we did not expect infection status to change during the experiment due to the isolation of individual oysters, during immersion when *P. marinus* transmission occurs, we focused on the effects of air temperature on infection intensity. The RFTM methodology provides an interpretable, quantitative assessment of intensity that can easily be compared with previous studies. A detailed explanation for the RFTM method can be found in *Malek &*

*Byers (2017)*. It is modified in the current study only by the collection of gill and mantle tissue in addition to rectal tissue for assessment because this increases the probability of detecting low intensity infections. Briefly, tissue was incubated in RFTM for six days, stained with Lugol's Iodine, and viewed under a dissecting microscope to score the abundance (intensity) of live *P. marinus* hypnospores, which enlarge and become visible microscopically during incubation. We found during the experiment that this methodology was likely limited in detecting infections at high temperatures ($\geq$43°C) as we observed lower infection prevalence in high ($\geq$43°C) compared to the low ($\leq$39 °C) temperature treatments, which were more comparable to our initial baseline of 60%. Because the RFTM method is designed for the detection of live parasites, if parasite mortality or dormancy occurred at higher temperatures, we would not be able to detect the parasite within the host, thus potentially accounting for the lower prevalence that was observed. Thus, for our analyses that included *P. marinus* intensity (see below), we excluded the uppermost temperature treatments, and focused only on the range from 27–39 °C.

To determine how oyster survival responds to increasing temperature and if survival was affected by the interaction of temperature and the intensity of *P. marinus* infection, we used the 'Survival' package in R (*Therneau, 2015*), which utilizes Kaplan–Meier estimation to evaluate survival and generate survival curves. We first analyzed survival by air temperature using the full temperature range, 27–53 °C, and next analyzed survival as a function of temperature (27–39 °C) and *P. marinus* infection intensity (RFTM Mackin score for 27–39 °C), and their interaction. As aforementioned, we used a truncated temperature range for the latter analysis to account for reliability issues of RFTM at high temperatures.

In addition to testing for effects of *P. marinus* infection intensity on oyster survival, we also wanted to know if the parasite itself was affected by increasing air temperature. We ran a separate analysis of infection intensity as a function of air temperature using a one-way analysis of variance (ANOVA), using RFTM data for the 27–39 °C treatments. If air temperature was a significant predictor of *P. marinus* infection intensity, we conducted post-hoc Tukey's HSD pairwise comparisons to identify differences between air temperature treatments.

**Host immune response experiment**
We ran a second experiment with a new collection of individual oysters and the same experimental setup described above to test for effects of increasing air temperature on *C. virginica* immune response. Because immune response and survival may be linked, and for comparability with survival data, we chose to again use a similarly broad and high range of air temperatures. However, to achieve the best resolution and boost sample sizes, we eliminated the 31 and 53 °C treatments used in the previous experiment. We used 25 individuals per treatment except for higher air temperatures (43–50 °C) which had 35 individuals per treatment to try to increase the number of days that live oysters could be sampled from those treatments (because we knew the mortality rate would be high). Four times over the course of two weeks (at two, four, seven and 14 d), we randomly selected six live individuals from each air temperature treatment for host immune response assessment. Several high air temperature treatments experienced complete oyster mortality

within several days (43–50 °C), so only the 27–39 °C treatments could be sampled at all four time points. The exact start time of air exposure was slightly staggered on sampling days so that individuals could be sampled immediately after removal from air temperature chambers.

We used phagocytic activity, the primary means of protection against foreign cells, as a measure of host immune response. Phagocytic activity was assessed by using fluorescent latex beads as a proxy for foreign cells. We measured phagocytic consumption of beads by granulocytes (the largest and most phagocytically active oyster hemocytes) in oyster hemolymph with flow cytometry following the methods of *Goedken & De Guise (2004)* and instruction of M. Levin (University of Connecticut, pers. comm., 2013). From this assessment, we were able to calculate the proportion of highly active granulocytes (those that consumed ≥3 beads; hereafter, 'highly active cells') and the mean number of beads consumed by all granulocytes (hereafter, 'mean beads consumed') in each air temperature treatment. See *Malek & Byers (2016)* for detailed flow cytometry methods.

Because granulocytes are the first line of defense for the oyster immune response, we also tested for effects of *P. marinus* infection status on phagocytic activity to better understand the relationship between the above-measured host immune response and parasite infection for each oyster. Because of previously discussed issues with *P. marinus* detection at high temperatures using the RFTM method, and also because we were interested in looking at the effects of the presence of the parasite on hemocyte activity (*Anderson, Burreson & Paynter, 1995*), it was necessary to use a more sensitive assay (qPCR) to detect infections (*Gauthier, Miller & Wilbur, 2006*; N Stokes, pers. comm., 2013). This methodology enabled us to detect any form of parasite DNA (live, dead, or degraded) and we could evaluate *P. marinus* infection status across all temperatures. Detailed methods for qPCR assessment, adapted from *Gauthier, Miller & Wilbur (2006)* and N Stokes (pers. comm., 2013), can be found in *Malek & Byers (2016)*.

To determine if increasing temperature affected the behavior of the highly active cells and the mean beads consumed within the host, and whether these traits were affected by *P. marinus* infection, we used a series of ANOVAs, one for each sampling day (2, 4, 7, 14), that included air temperature and *P. marinus* infection status as fixed effects. If air temperature was a significant predictor of phagocytic activity, we conducted post-hoc Tukey's HSD pairwise comparisons to identify differences between air temperature treatments. Each measure of phagocytic activity (the proportion of highly active cells and mean beads consumed) was analyzed with separate ANOVAs. We arcsine square root transformed the proportion of highly active cells (consuming ≥3 beads) to meet the assumptions of normality. As with the Host Survival Experiment, the 27 °C subtidal treatment was not included in formal statistical analyses. All analyses were run in R version 3.2.0 (*R Core Team, 2015*).

## RESULTS

### Host survival experiment

There was no interactive effect of air temperature (27–39 °C) and *P. marinus* infection intensity as measured by the RFTM method on *C. virginica* survival, but there was a
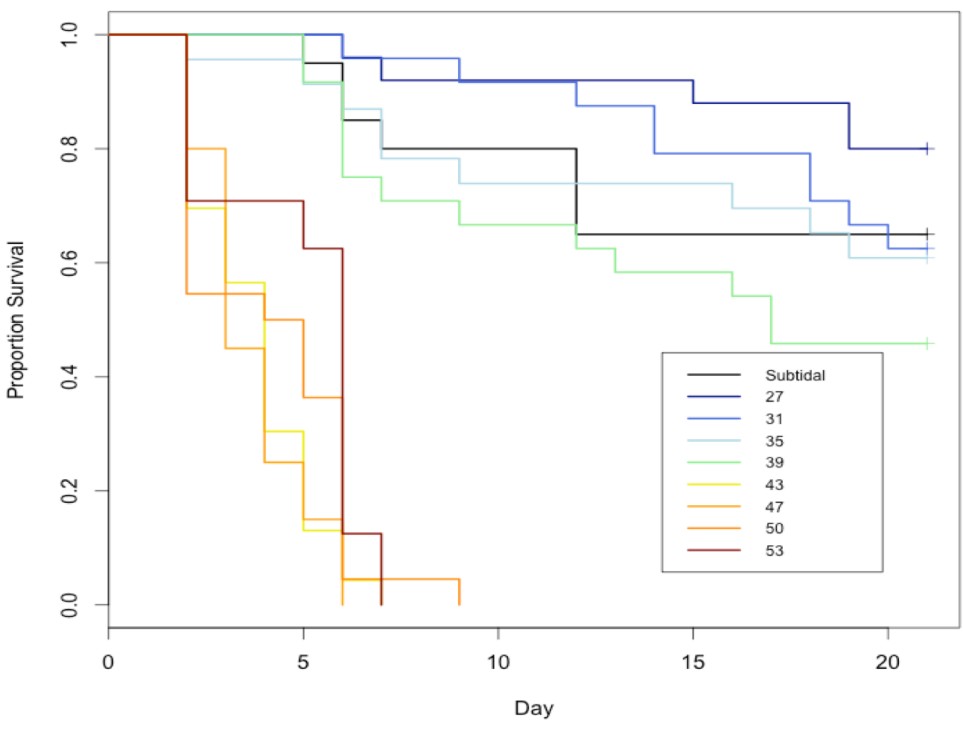

**Figure 2** Total survival of all experimental oysters by air temperature, regardless of infection status.
$n \approx 25$ for each air temperature curve.

**Table 1** Results of Kaplan Meier analysis of *C. virginica* survival in the Host Survival Experiment, based on a. air temperature and b. air temperature and *P. marinus* infection intensity.

| Source | Coef | Exp (coef) | SE (coef) | z | p (z) |
|---|---|---|---|---|---|
| a. Survival by air temperature treatment (27–53 °C) | | | | | |
| Air temperature | 0.216 | 1.242 | 0.019 | 11.6 | <0.001 |
| b. Survival by air temperature treatment and *P. marinus* infection intensity (27–39 °C) | | | | | |
| Air temperature | 0.112 | 1.118 | 0.050 | 2.217 | 0.027 |
| Infection intensity | 0.552 | 1.737 | 1.342 | 0.412 | 0.681 |
| Temperature*Intensity | −0.013 | 0.987 | 0.039 | −0.346 | 0.729 |

**Notes.**
Infection intensity is based on RFTM assessment.

significant effect of air temperature (Fig. 2, Table 1). We observed a clear delineation in response to air temperature between 39 and 43 °C. Air temperature treatments ≤39 °C maintained at least 50% oyster survival over 21 days and treatments ≥43 °C reached 0% oyster survival within several days (Fig. 2). There was no effect of *P. marinus* intensity on survival (Table 1B). Because we also observed complete host mortality within several days at temperatures above 39 °C (Fig. 2), the truncated temperature range over which *P. marinus* infection intensity was evaluated provided more ecologically relevant insights on the relationship between host and parasite.

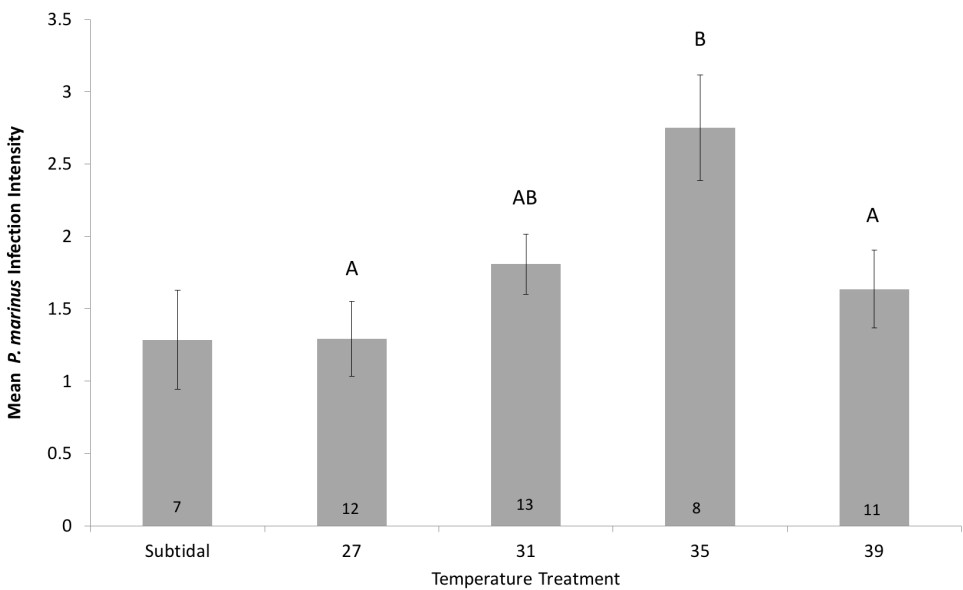

**Figure 3** **Mean *P. marinus* infection intensity (based on RFTM assessment) by air temperature treatment (27–39 °C) in the Host Survival Experiment.** Mean intensity was calculated using only infected oysters (both live and dead) and the numbers inside the bars indicate the number of infected oysters used to calculate mean intensity for each treatment. Standard error bars were calculated across individual oysters within each treatment. Capital letters indicate results of post hoc Tukey's HSD analysis; different letters indicate statistically significant differences between treatments and similar letters indicate no statistical difference between treatments. The subtidal treatment is included in the figure for comparison, but was not part of the formal analysis.

**Table 2** **Results of 1-way ANOVA testing for the effect of air temperature (27–39 °C) on *P. marinus* infection intensity in the Host Survival Experiment.**

| *P. marinus* infection intensity by air temperature (27–39 °C) and survival status | | | | | |
| Source | Df | Sum of squares | Mean square | F value | P(F) |
| --- | --- | --- | --- | --- | --- |
| Air temperature | 3 | 10.615 | 3.538 | 4.522 | 0.009 |

**Notes.**
Infection intensity is based on RFTM assessment.

Analysis of variance indicated that air temperature also had a significant effect on infection intensity (Fig. 3, Table 2). Tukey's HSD pairwise comparisons indicated that oysters exposed to 35 °C had significantly higher infection intensity than oysters exposed to other temperatures, with the exception of 31 °C (Fig. 3).

## Host immune response experiment

There was a significant effect of air temperature on both measures of phagocytic activity, but only on sampling day 2, which was the one day that had equal sample sizes for all temperature treatments (highly active cells $df = 5$, $F = 3.084$, $p = 0.022$; mean beads consumed $df = 5$, $F = 4.103$, $p = 0.006$) (Fig. 4). Air temperature did not affect phagocytic activity on the other sampling days ($p > 0.05$). Tukey's HSD comparisons on day 2 indicated that the there was a significantly lower proportion of highly active cells at 47 ($p = 0.046$)

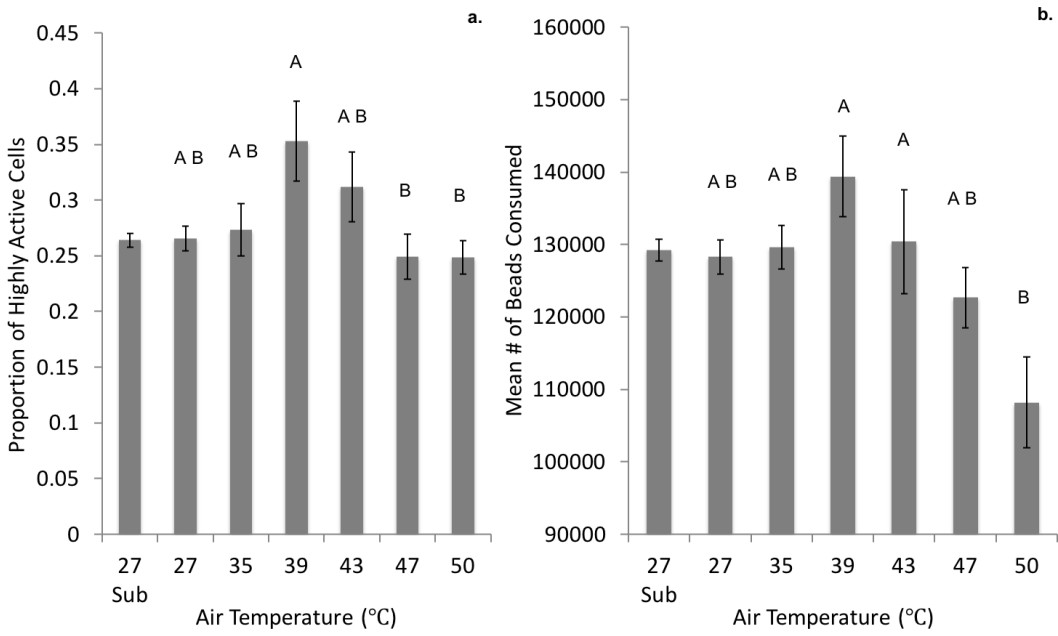

**Figure 4** Phagocytic activity of granular oyster hemocytes (granulocytes) in the Host Immune Response Experiment on sampling day 2. (A) the mean proportion of highly active cells (indexed by the consumption of ≥3 fluorescent beads; left panel) and (B) the mean number of beads consumed (average beads consumed by all oyster granulocytes; right panel) in response to air temperature. Error bars represent standard error calculated across individual oysters within each air temperature treatment ($n = 6$ for each treatment). Capital letters indicate results of post hoc Tukey's HSD analysis; different letters indicate statistically significant differences between treatments and similar letters indicate no statistical difference between treatments.

and 50 °C ($p = 0.029$) compared to 39 °C. Significantly fewer beads were consumed at 50 °C compared to 39 ($p = 0.002$) and 43 °C ($p = 0.050$). There was no effect of *P. marinus* infection status on either measure of activity ($p > 0.05$ for all analyses).

## DISCUSSION

Climate change is predicted to affect environmental factors that shape host-parasite interactions. We found that increasing air temperature can affect the traits of the oyster *C. virginica* and infection intensity patterns of its parasite, *P. marinus*. The temperatures at which we observed changes in host traits (survival) and parasite infection patterns (intensity) are within the range of air temperatures projected to occur in the Southeast US within the next one hundred years (*Mitchum, 2011*). Specifically, our results indicated that host survival decreases as temperatures move towards 35 °C, then drop quickly above 39 °C. Phagocytic activity of oyster hemocytes however was largely unaffected until at least 47 °C, at which point the host experienced complete mortality, suggesting that increasing temperature is unlikely to affect granulocyte activity at temperatures that are physiologically favorable for the host. Infections intensity of *P. marinus* increased with temperature, peaking at 35 °C, and then declined significantly at 39 °C. Even though significantly lower compared to 35 °C, intensity at 39 °C was comparable to the intensity

observed at previous temperatures (31 °C), suggesting that the parasite may be more tolerant of increases in temperature than its host, at least within the range tested (Fig. 3). As temperatures rise in the intertidal, oysters may suffer not only from temperature-related mortality in the short-term (weeks), but also from higher intensity *P. marinus* infections over the long-term (months).

The current maximum average intertidal air temperature in mid-coast Georgia in the summer is ∼33 °C, therefore increasing average air temperature in this region, even by several degrees, could cause an increase in *P. marinus* infection intensity. The higher intensity parasite infections observed at 35 °C may be detrimental to the oyster host because intensities greater than 2 on the Mackin scale are more likely to cause harm to the host through effects such as decreased shell and soft tissue growth (*Ray, Mackin & Boswell, 1953*; *Menzel & Hopkins, 1955*), reduced reproductive development (*Dittman, 1993*), and weakened adductor muscles (*Mackin, 1962*). Over time, these effects can lead to host mortality. We did not find a significant relationship between host survival and parasite infection intensity (Table 1). However, the relatively short duration of our experiment likely prevented us from detecting negative effects of the higher intensity infections, which can take months to over a year to cause host mortality, depending on ambient water temperature and salinity conditions (*Mackin, 1951*).

Interestingly, we did not see an effect of air temperature on phagocytic activity, except at very high temperatures. Although oyster hemocytes have been shown to be sensitive to environmental factors such as temperature, salinity, air exposure, and parasite infection (*Fisher, Chintala & Moline, 1989*; *Anderson, Paynter & Burreson, 1995*; *Hégaret, Wikfors & Soudant, 2003*; *Duchemin, Fournier & Auffret, 2007*; *Kuchel et al., 2010*), our results suggest that air temperature only affected hemocyte activity at temperatures that cause rapid oyster mortality. These effects of temperature were only analyzable across the full temperature range on day 2, which was the single sampling day on which we had samples from all temperatures due to high oyster mortality above 39 °C. Thus, any continuation of the pattern of decreased phagocytic activity at these temperatures over time would not be ecologically significant as the oyster would die from temperature exposure before any adverse impacts of decreased phagocytic activity occurred.

Hemocyte characteristics can be sensitive to parasite infections (*Anderson, Burreson & Paynter, 1995*) and *P. marinus* in particular can use host phagocytic activity as a means to facilitate entry into host hemocytes (*Tasumi & Vasta, 2007*). However, our results indicated no effect of *P. marinus* infection status on phagocytic activity. Similarly, *Keppel et al. (2015)* found that *P. marinus* intensity scores did not correlate with hemocyte activity. The presence of the parasite alone, which can affect hemocyte number (*Anderson, Paynter & Burreson, 1995*), may not influence the activity of hemocytes, and other immune functions (i.e., apoptosis) may be more important for regulating or responding to *P. marinus* infection (*Anderson, Paynter & Burreson, 1995*; *Goedken et al., 2005*).

## CONCLUSIONS

There is much speculation about how changes in climate will affect host-parasite interactions (*Harvell et al., 2009*; *Lafferty, 2009*). Our study suggests, at least for oysters

and one of their most detrimental parasites, that a temperature increase of just a few degrees above the current intertidal average in Georgia could affect host survival and lead to higher intensity *P. marinus* infections. Similar to air temperature, water temperature is also predicted to increase (*IPCC, 2014a*), and may have similar effects on *P. marinus* infections as the parasite grows and reproduces more quickly at warmer water temperatures (*Andrews, 1988*; *Chu & Volety, 1997*). Though we looked at air temperature in isolation, global environmental change likely will result in concomitant changes in water and air temperature, and other factors (e.g., water acidity, salinity, oxygen concentrations), that may have interactive effects that could further influence host and parasite characteristics and interactions. Observed changes in *C. virginica* populations in the future may more likely be temperature-related over the short-term as temperature increases but be more parasite-driven as higher intensity infections, influenced by increased temperature, begin to impact the host over the longer term. Recognizing that host and parasite responses to increasing temperatures differ and may be variable across temporal scales will help us to more effectively shape our future research initiatives for other ecologically and economically valuable species, and develop a stronger understanding of the relationship between changes in climate, hosts, and their parasites.

## ACKNOWLEDGEMENTS

The authors thank the University of Georgia Marine Extension Shellfish Research Lab for providing a temperature-controlled space to complete these experiments, the UGA Skidaway Institute of Oceanography for field site access. We would also like to especially thank Camden Lowrance for developing the code to run the air temperature chambers, as well as Daniel Harris, Chris Malek, and Ashton Potter who assisted with the electrical and technical setup of the Arduino system. The Byers' lab and Kristy McDowell provided significant feedback on this manuscript and Morgan Walker provided substantial assistance in experimental setup and oyster processing.

### Funding

This work was supported by the Odum School of Ecology's Small Grants program, University of Georgia. The funders had no role in study design, data collection and analysis, decision to publish, or preparation of the manuscript.

### Grant Disclosures

The following grant information was disclosed by the authors:
Odum School of Ecology's Small Grants program,.

### Competing Interests

The authors declare there are no competing interests.

## Author Contributions

- Jennafer C. Malek conceived and designed the experiments, performed the experiments, analyzed the data, contributed reagents/materials/analysis tools, prepared figures and/or tables, authored or reviewed drafts of the paper, approved the final draft.
- James E. Byers conceived and designed the experiments, analyzed the data, contributed reagents/materials/analysis tools, prepared figures and/or tables, authored or reviewed drafts of the paper, approved the final draft.

## Data Availability

The raw data are provided in a Supplemental File.

## Supplemental Information

Supplemental information for this article can be found online at http://dx.doi.org/10.7717/peerj.5046#supplemental-information.

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
