# Peer review of "Responses of an oyster host (Crassostrea virginica) and its protozoan parasite (Perkinsus marinus) to increasing air temperature"

_PeerJ, doi:10.7717/peerj.5046_

## Round 0.1 · original submission · Minor Revisions

Overall the manuscript is well written, with solid experimental design and appropriate inference drawn from the data.

The reviewers point out a few areas of improvement that should be addressed in the revised manuscript.

In addition, it would be helpful if hemocytes and granulocytes were defined (are they the same cells?). Some discussion about the likely co-variance of food availability/quality with temperature is also warranted, given warming is occurring alongside other environmental changes (Dore, Lukas, Sadler, Church, & Karl, 2009; IPCC Summary for Policymakers, 2014), and the strong likelihood of interactive effects.

·

Basic reporting

This manuscript explores the effect of increasing air temperatures on the interaction of the eastern oyster (Crassostrea virginica) with the protistan parasite Perkinsus marinus. The number of infected oysters, the parasite load and oyster immune responses were determined. In addition, oyster mortality was monitored. Results show that at temperatures above 35°C, host survival decreases and oyster hemocytes are not affected until lethal temperatures are reached. In addition, P. marinus infection intensity increased at 35°C compared to the other temperatures.

The language is clear and unambiguous, the introduction and background are appropriate and the literature is cited appropriately. All figures and tables are relevant and well labelled. Raw data is supplied with the exception of temperature variation in the exposure chambers.

Experimental design

The research question is well defined and relevant. Methods are described in sufficient detail but see following comments. The experimental chamber description is hard to follow. Is it possible to provide a photo or diagram of the setup? Did the temperature vary more than 0.5°C? The use of a temperature data logger is indicated but there is no mention of the temperature variation in actual experiments.

Validity of the findings

The conclusions are valid, and data is robust. Conclusions are well stated and support the results.

Additional comments

The authors state that the average surface air temperature will increase from 0.3 to 4.5°C by 2090 and that the average in the area under study is 33°C. Why then were temperatures up to 53°C used for the experiments as this is well above the 37.5 projected temperature. The reasons for the specific temperatures chosen should be discussed.

The authors speculate that increased temperatures may in fact cause an increase in immune responses of the host. Isn’t it also the case that this increase in immune response is harmful in and of itself? Isn’t it possible that an increase in immune response damages the host more than it protects the host?

Line 321 – remove are considered

·

Basic reporting

The manuscript by Malek and Byers describes interesting results on the potential effects of rising atmospheric temperatures on host/parasite interactions and survival. The study will be of broad interest because of the not only because of the topic, but also because of the host is an economically and ecologically important species and the pathogen is a member of a genus that infects bivalves worldwide. The design and range of experiments, and the variety of methods including pcr, rftm and tests of phagocyte activity, is a strong point of the study. The ms is clear and well written. Figures and tables are appropriate. I recommend acceptance after very minor revision to address the issues raised below.

Experimental design

The design and analysis of experiments is very good. The variety of approaches to the problem is a strong point of the study. I have only 2 minor comments that should be addressed:

line 192. Add the explanation that using gill and mantel tissue in addition to rectal tissue increases the probability of detecting low-intensity infections.

line 199. Could Perkinsus be dormant – not dead? Omitting the highest temperature treatments would still be the appropriate step, but the implication could be different if infections could become active again as temperatures cooled in autumn.

Validity of the findings

The data are robust and the interpretation of the findings is sound. The conclusions are logical extensions of the results and are not overly speculative.

Some minor suggestions for the authors to consider:

For Perkinsus infections, increased phagocyte activity does not always correlate with decreasef infection prevalence (Keppel et al., 2015). Please include a comment on this.

Volety has worked on temperature effects on this host/parasite system. This is relevant even though it focused on subtidal oysters.

line 314. For the sentence: ‘As temperatures rise in the intertidal, oysters may suffer not only from temperature-related mortality in the short- term, but also from higher intensity P. marinus infections over the long-term.’ – please explain what is meant by short- and long-term a bit more clearly. Do you mean days (survival) to months (disease) or over decades as temperatures rise.

---

## Round 0.2 · accepted · Accept

Thank you for your revised manuscript. I noticed the legend for Table 1 and 2 was switched - this will require checking. Also, your schematic for the experimental setup would best identify the components with some text labels rather than explaining them in the legend. These things can be resolved while in production

#